# Lateral Evasive Maneuver with Shared Control Algorithm: A Simulator Study

Joseba Sarabia [1,2,*], Mauricio Marcano [1], Sergio Díaz [1], Asier Zubizarreta [2] and Joshué Pérez [1]

1 TECNALIA, Basque Research and Technology Alliance (BRTA), Astondoa Bidea, Edificio 700, 48160 Derio, Spain; mauricio.marcano@tecnalia.com (M.M.); sergio.diaz@tecnalia.com (S.D.); joshue.perez@tecnalia.com (J.P.)
2 Bilbao School of Engineering, University of the Basque Country UPV/EHU, 48013 Bilbao, Spain; asier.zubizarreta@ehu.eus
* Correspondence: joseba.sarabia@tecnalia.com or jsarabia003@ikasle.ehu.eus

**Abstract:** Shared control algorithms have emerged as a promising approach for enabling real-time driver automated system cooperation in automated vehicles. These algorithms allow human drivers to actively participate in the driving process while receiving continuous assistance from the automated system in specific scenarios. However, despite the theoretical benefits being analyzed in various works, further demonstrations of the effectiveness and user acceptance of these approaches in real-world scenarios are required due to the involvement of the human driver in the control loop. Given this perspective, this paper presents and analyzes the results of a simulator-based study conducted to evaluate a shared control algorithm for a critical lateral maneuver. The maneuver involves the automated system helping to avoid an oncoming motorcycle that enters the vehicle's lane. The study's goal is to assess the algorithm's performance, safety, and user acceptance within this specific scenario. For this purpose, objective measures, such as collision avoidance and lane departure prevention, as well as subjective measures related to the driver's sense of safety and comfort are studied. In addition, three levels of assistance (gentle, intermediate, and aggressive) are tested in two driver state conditions (focused and distracted). The findings have important implications for the development and execution of shared control algorithms, paving the way for their incorporation into actual vehicles.

**Keywords:** shared control; automated driving; driver-automation cooperation; simulator-based study; safety; user acceptance

## 1. Introduction

In recent years, the automotive industry has seen a considerable increase in the implementation of automated driving technologies due to their potential to improve road safety and efficiency [1]. These technologies range from Level 3 to Level 5 on the SAEJ3016 automation scale [2], and they allow sharing or even overriding of vehicle control from the human driver.

However, despite the increase in the number of works and developments related to highly automated systems for vehicles, completely replacing human drivers with these approaches remains a challenging task [3,4]. In fact, human drivers are still the most dependable agents when performing a dynamic driving task, with safety statistics still surpassing those of automated systems when performing the driving task [5]. Nevertheless, their performance decreases significantly when their role is to oversee the automated system and they do not take part actively in the driving task [6].

Moreover, the legislation issues related to the allocation of faults or damages related to a crash or accident is still a challenging issue when automated driving technologies are involved. A recent study [6] has revealed that in scenarios where the driver can override the automated system for corrective action (such as cases in which the driver can operate the

steering wheel or the pedals), the blame of the accident falls on the human driver and not on the automated system. This study further highlights that even if it can be demonstrated that human skills in monitoring and supervising the automated system are sub-optimal and fail to meet safety standards compared to when the human driver is solely in control, the responsibility for crashes continues to rest with the human driver.

The aforementioned issues have led some authors to propose cooperation between human drivers and vehicle automated systems during the driving task as a means to improve safety and efficiency [7–9]. Among the proposed approaches to overcome this challenge, shared control algorithms have emerged as a potential solution, as these algorithms allow active driver engagement in the driving task while maintaining continuous control assistance through the automated system [10].

One of the typical applications of shared control is its implementation as part of the lateral control of the vehicle by means of a haptic control strategy that allows both the human driver and the automated system to interact with the steering wheel [11]. This approach allows a combination of the torque generated by both agents, allowing mutual intentions to be perceived and predicted [12,13].

Based on the input torque applied by the driver on the steering wheel, some of the works proposed in this area have focused on predicting the driver's intention to manage the driver-automated system interaction. Refs. [14,15], for instance, use the applied torque as a reference for the trajectory planner. Other works, such as [16], use the lateral offset and the lateral velocity, while [17] also considers the angular deviation caused by the driver to modify the trajectory. These approaches are cost-effective, as they require no additional in-vehicle equipment. However, predicting the driver's intention is a complex task, generally applied to easy driving conditions. For safety-critical situations, on the contrary, human response is less predictable, and the previous applications can no longer be applicable.

In safety-critical situations, rather than predicting the driver's intention, the automated system is authorized to actively participate in the control loop, compensating or correcting the driver to prevent vehicle off-road incidents or collisions. This is the case, for instance, in lane-keeping systems, which aim to aid drivers in maintaining their position within the lane, particularly during moments of distraction. Several studies have studied the application of shared control for these systems [18,19], with a range of steering support torque between 2–4 Nm. A particularly interesting work in this field is that of Park et al. [20], who focused their research on determining the optimal steering torque for lane-centering support, finding an average optimal value of 2.5 Nm based on objective and subjective metrics.

However, the aforementioned works are not focused on critical maneuvers, in which the assistance levels should be higher. Common critical scenarios often involve unsafe lane changes, often due to factors such as vehicles appearing in blind spots or unintended road departures. Steering assistance in these scenarios has been explored at varying torque levels, such as 5 Nm [21,22], 8 Nm [23], and even close to 10 Nm [24,25]. These later studies have shown promising results in mitigating most unsafe incidents.

This work aims to further contribute to this latter field by introducing an even more critical maneuver, namely lane invasion, where the steering support level is evaluated in a thorough study considering a balance between safety and driver acceptance. The proposed work presents a similar approach to the overriding Guardian Angel concept presented in [26]. In particular, the focus of the developed approach focus lies on preventing frontal collisions caused by lane invasions.

To conduct this study, a combination of objective indicators—such as successful collision avoidance and avoidance of off-road incidents—and subjective indicators—capturing drivers' perceptions of the maneuver—were incorporated. Additionally, given the pivotal role of driver-automated system interaction in shared control, an assessment of user acceptance was undertaken by comparing three distinct levels of driving assistance torques.

This work has been performed under the umbrella of the HADRIAN project https://hadrianproject.eu/ (accessed on 4 May 2023) and the Aware2all project https://www.

ccam.eu/projects/aware2all/ (accessed on 15 October 2023), funded both by the European Commission. These projects are focused on the improvement of human-to-vehicle interaction in the scope of automated driving features. For that, different interaction channels are explored, and this work focuses on interaction based on the direct collaboration at the steering wheel during potential collision situations.

The rest of the paper is organized as follows: Section 2 describes the setup of the experiment and provides a comprehensive explanation of the developed controller. Section 3 describes the test procedure, along with the study design and the defined metrics. In Section 4, the results obtained from the experiment are presented and distributed into objective and subjective metrics. Section 5 is dedicated to the discussion of these results, providing in-depth analysis and interpretation. Lastly, in Section 6, the key findings of the study are summarized, and the concluding remarks are presented.

## 2. Experimental Setup

### 2.1. Test Scenario

The statistics from the Spanish National Traffic Organization (DGT) indicate that 75–80% of total road casualties occur on two-way conventional roads with speed limits up to 90 kph. Of these cases, 40% are related to vehicles going off-road, while 27% are casualties derived from frontal collisions [27].

Hence, this section proposes a haptic shared control approach focused on preventing collisions in two-way conventional roads. The proposed test scenario is designed to force an emergency lateral evasive maneuver to prevent a frontal collision and the following off-road situation.

In this test scenario, the ego vehicle (depicted in blue) travels at 90 kph along a two-way road. Meanwhile, on the opposite lane, an oncoming vehicle (depicted in brown), also at 90 km/h, and a motorcycle are approaching. The oncoming vehicle blocks the visibility of the motorcycle behind. Suddenly, the motorcyclist performs a reckless overtake, invading the ego vehicle's lane and leaving little time to react and maneuver to avoid a collision, as the ego vehicle has no time to anticipate the situation. Figure 1 depicts the maneuver sequence.

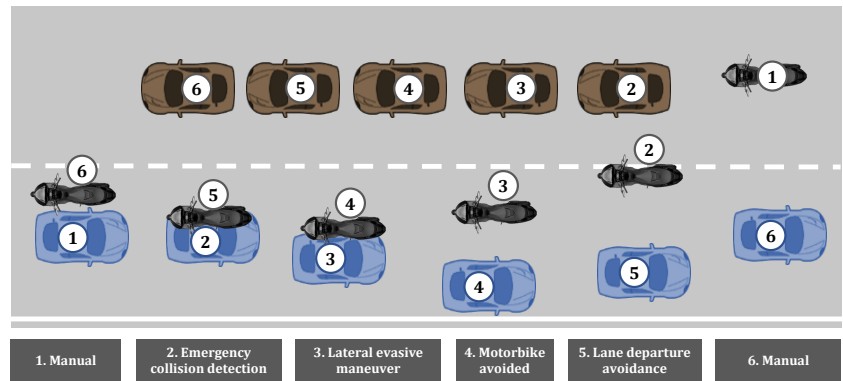

**Figure 1.** Lateral evasive maneuver sequence.

To provide the same reaction time before the potential crash, the motorcycle is programmed to appear at a specific distance from the ego vehicle. In addition, the motorcycle does not perform the overtake maneuver in every scenario in order to prevent the participants from predicting this scenario.

Both crashing into the motorcycle and departing the lane (going off-road) are considered to be an accident. Consequently, the space allowed for a safe maneuver is defined as shown in Figure 2.

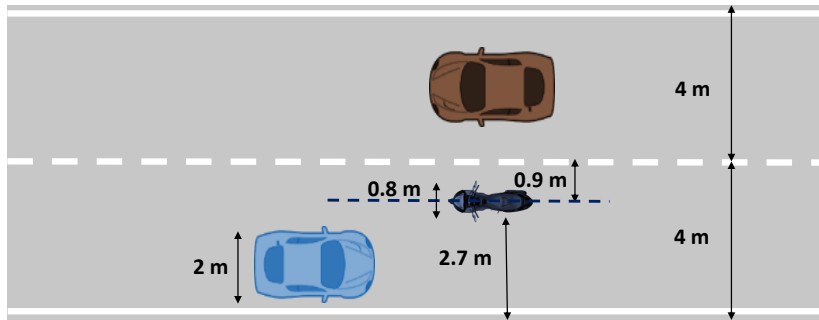

**Figure 2.** Safe maneuver scenario.

## 2.2. Setup and Equipment

To perform the simulator-based study, a simulation-based environment is used (Figure 3). The simulator is developed using several off-the-shelf components that are integrated to provide a realistic driving experience. The steering wheel is part of the Augury H kit, which consists of a servomotor actuator, a Simucube motor controller, and a racing steering wheel. The servomotor actuator includes a brushless DC motor with a current sensor and an incremental encoder. The actuator can generate a maximum torque of 18 Nm and a power of 1.5 KW. The applied torque is measured through the current sensor. The simulator is complemented by racing pedals featuring mechanical damping and a racing seat.

The developed simulator includes three 32-inch LCD screens that allow realistic immersion in the driving task by using rendered 3D environments of the scenario. In addition, a touch display has been included to emulate a phone and analyze the effect of performing secondary tasks while driving. The vehicle dynamics and the proposed scenario are simulated in Dynacar® [28] while the required control algorithms are executed in Matlab Simulink R2021b .

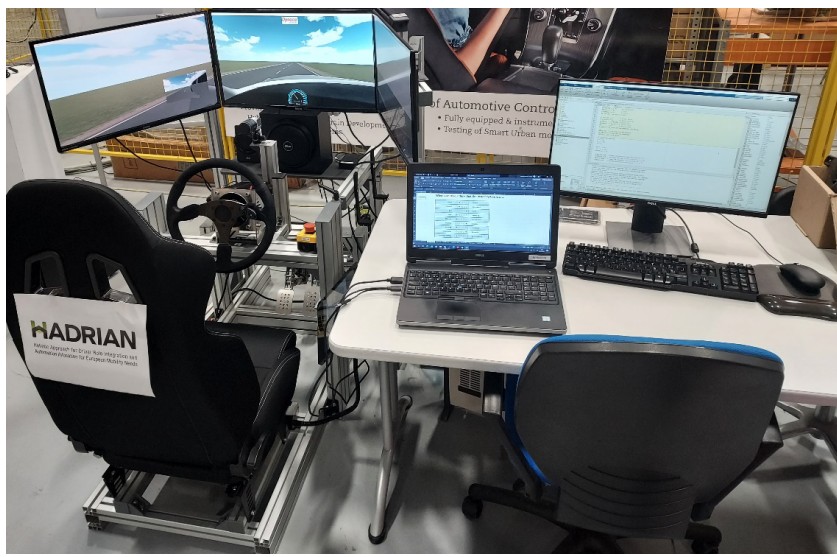

**Figure 3.** Driving simulator setup for the user tests.

## 2.3. Shared Controller

The haptic shared controller used in this work is based on the one proposed in [19]. This controller is built upon three main principles:

1. **It applies torque as the control signal** to seamlessly cooperate with the driver, instead of treating the driver as a disturbance (as position-based controllers do [29]).

2. **The controller has the ability to adjust its authority** (i.e., stiffness), thus providing varying levels of resistance to the driver's steering wheel actions. Also, it is essential for the controller to maintain stability across different values of authority.

3. **The controller relies on optimal control** to balance multiple objectives within the driver-automation system (e.g., comfort, safety, and effectiveness) [10].

The shared control approach is implemented using nonlinear model predictive control (NMPC), known for effectively managing nonlinearities and constraints while optimizing performance over time horizons, leading to superior control performance compared to approaches that consider only the current system state.

This control approach requires the definition of a dynamic model that comprises the vehicle dynamics, the road–vehicle representation, and the steering wheel model. The dynamic equations can be rewritten as a nonlinear state-space function [19],

$$\dot{\mathbf{x}}(t) = \mathbf{f}(\mathbf{x}(t), \mathbf{u}(t)) \tag{1}$$

where $\mathbf{x}(t)$ is the state-space vector, comprised of the variables that define the dynamic model, such as the vehicle's position and orientation in the global frame $[X, Y, \Psi]$, longitudinal and lateral speeds $[\dot{v}_x, \dot{v}_y]$, yaw rate $\dot{\Psi}$, wheel turn angle $\delta$, lateral and angular errors $[e_y, e_\Psi]$, and steering wheel angle and rotational speed $[\theta, \omega]$. On the other hand, $\mathbf{u}(t)$ represents the control action sequence to be calculated by the NMPC controller, in which the control torque exerted by the actuated steering wheel $T_{mpc}$ and its discrete-time derivative $\Delta T_{mpc}$ are included.

This way, the NMPC control law can be defined as a constrained optimization problem $V$ that can be solved using an NMPC solver, as detailed in [19],

$$\min_{T_{\mathrm{mpc}}} \quad J(\mathbf{x}(\mathbf{k}), \mathbf{u}(\mathbf{k})) \tag{2}$$

s.t.

$$|T_{\mathrm{mpc}}(k)| \quad \leq T_{\max}, \quad k = 1, \dots, N \tag{3}$$

$$|\Delta T_{\mathrm{mpc}}(k)| \leq \Delta T_{\max}, \quad k = 1, \dots, N \tag{4}$$

$$|e_y(k)| \quad \leq e_{y_{\max}}, \quad k = 1, \dots, N \tag{5}$$

$$|\psi(k)| \quad \leq \psi_{\max}, \quad k = 1, \dots, N \tag{6}$$

where $k$ is the time step after discretization and $N$ is the prediction horizon of the NMPC.

The cost function $J(\mathbf{x}(\mathbf{k}), \mathbf{u}(\mathbf{k}))$ is defined to allow tracking the desired vehicle trajectory ($[X, Y, \Psi]$) and reducing the drift related to the yaw rate ($\psi$) during the evasive maneuver, while minimizing the control torque ($T_{mpc}$) and its derivative ($\Delta T_{mpc}$) in order to improve the comfort of the steering assistance,

$$J(\mathbf{x}(\mathbf{k}), \mathbf{u}(\mathbf{k})) = ||X - X_r||^2_{W_X} + ||Y - Y_r||^2_{W_Y} + ||\Psi - \Psi_r||^2_{W_\Psi} + \tag{7}$$

$$||\psi - \psi_r||^2_{W_\psi} + ||T_{mpc}||^2_{W_T} + ||\Delta T_{mpc}||^2_{W_{\Delta T}} \tag{8}$$

where $\mathbf{W_X}, \mathbf{W_Y}, \mathbf{W_\Psi}, \mathbf{W_\Psi}, \mathbf{W_T}$ and $\mathbf{W_{\Delta T}}$ are the corresponding optimization weight matrices.

The constraints have been defined to ensure proper operation of the shared control. Safety-related constraints have been included to ensure that the calculated torque prevents vehicle deviation from the road ($e_{y_{max}}$ and $\psi_{max}$), resulting in an off-road scenario during the evasive maneuver.

In addition, torque constraints have been included to prevent discomfort in the driver-automated system interaction by limiting the actuated torque $T_{max}$ and its time change $\Delta T_{max}$. In fact, these variables are directly related to the stiffness of the controller (or authority), which is one of the main features of the shared controller proposed in [19].

For this purpose, an authority variable $\lambda$ is defined to adjust the stiffness of the controller and is included in the torque equation of the previously defined model as follows:

$$\dot{T}_{mpc} = \lambda \Delta T_{mpc} \tag{9}$$

The authority parameter $\lambda$ lacks units, making it difficult to intuitively assign its value. To address this problem, a new authority parameter, $\hat{\lambda}$, is introduced, which is defined as the maximum torque $\hat{\lambda} = T_{max}$. This torque is defined as the maximum experienced when approaching the boundary of the lane (specifically, at a distance of 2 m from the center of the lane).

The relationship between both parameters is related by a linear function which considers the physical torque limits of the steering wheel motor (15 N in the simulator used) and is adjusted experimentally,

$$\lambda = f(\hat{\lambda}) = \begin{cases} 2.4\hat{\lambda} - 6.3 & \text{if } \hat{\lambda} \geq 3 \\ 1 & \text{if } \hat{\lambda} < 3 \end{cases} \tag{10}$$

This way, for the test scenario and simulation equipment used, the NMPC has been tuned with a nominal value of $\hat{\lambda} = 3$, which limits the assistance to 3 N·m.

Note that as $\lambda$ increases, so does the stiffness of the controller, which may cause oscillations. Hence, as detailed in [19,23], a scaled damping variable $\hat{b}$ is introduced in the steering actuator motor controller to improve stability,

$$\hat{b} = b \sqrt{\frac{\lambda + 1}{2}} \tag{11}$$

where $b = 0.65$ N·m/s is the damping related to the steering system dynamics.

Finally, to perform the avoidance maneuver, the reference trajectory for the NMPC lateral controller is adjusted by considering a prediction horizon with a length of 1.5 s ($N = 30$). It is to be noted that the vehicle calculates two routes simultaneously, a reference route related to the regular driving ($r_A$) and the evasion reference route, which, as depicted in Figure 4, is located displaced to the right border of the lane ($r_B$). This way, if the relative distance $d(k)$ from a predicted future position of the vehicle and the motorcycle falls below a predefined threshold, the reference route changes from the regular one ($r_A$) to the evasion reference one ($r_B$).

The controller parameters, which have been tuned experimentally, are summarized in Table 1.

**Table 1.** Parameters of the NMPC lateral vehicle controller.

| **Vectors** | | | | | | *x* | | | | | *u* | *Δu* |
|---|---|---|---|---|---|---|---|---|---|---|---|---|
| **Variables** | **X** | **Y** | **Ψ** | $v_x$ | $v_y$ | $\psi$ | $e_y$ | $e_\psi$ | $\theta$ | $w$ | $T_{mpc}$ | $\Delta T_{mpc}$ |
| $W_z$ | 50 | 50 | 50 | 0 | 0 | 100 | 0 | 0 | 0 | 0 | 0.2 | 0.2 |
| Min/Max | - | - | - | - | - | ±0.75 | ±2 | - | - | - | ±$\hat{\lambda}$ | ±2$\lambda$ |
| Units | m | m | rad | m/s | m/s | rad/s | m | rad | rad | rad/s | N·m | N·m/s |

To illustrate the controller's performance, data from an evasive maneuver conducted with a distracted driver across various authority levels (represented by three distinct $\hat{\lambda}$ values) is presented below, in Figure 4. The results indicate consistent adherence to NMPC constraints and the vehicle's sustained performance and stability prior to, during, and after the maneuver. As expected, a higher $\hat{\lambda}$ value appears to enhance the likelihood of preventing a collision with the motorcycle. Note that the effect and a thorough analysis of the different authority values will be further explained next.

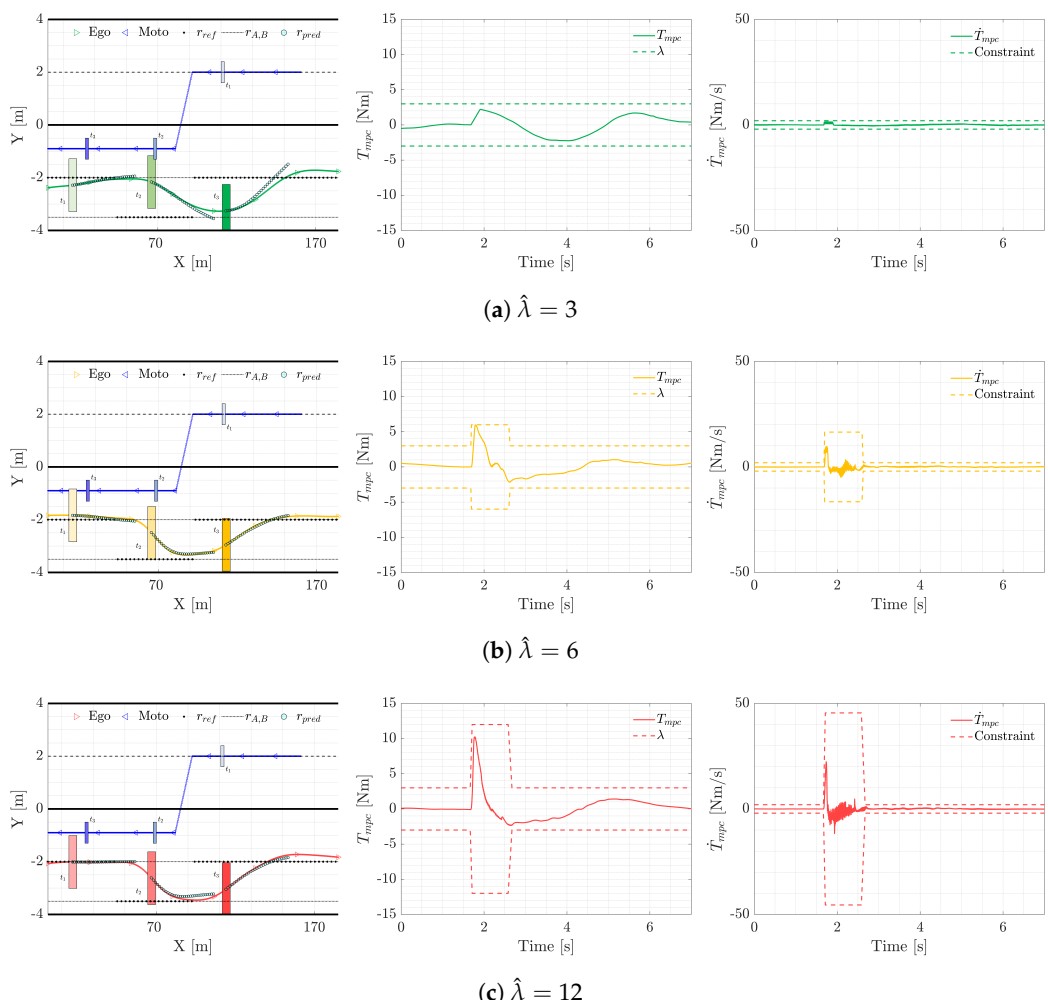

**(a)** $\hat{\lambda} = 3$

**(b)** $\hat{\lambda} = 6$

**(c)** $\hat{\lambda} = 12$

**Figure 4.** Ego vehicle trajectory, torque, and torque derivative responses to the lateral evasive maneuver for three different values of authority.

### 2.4. Test Configurations

In the proposed study, the performance of the previously detailed shared control approach for the lateral assistance maneuver detailed in Section 2.1 is evaluated. This evaluation encompasses various levels of driving assistance torques and two driver awareness levels.

Regarding the driver's state, two states are evaluated: attentive and distracted. The attentive state represents a normal driving scenario where the driver is attentive to the road. In this state, the automated system executes the longitudinal control via cruise control (CC) for speed regulation and a lane departure avoidance system for lateral control, where the driver can move freely inside the lane. Conversely, the distracted state simulates inappropriate driver behavior, where the driver is engaged in a secondary task rather than focusing on the road. Here, CC longitudinal control is combined with a lane-centering system [19] to enhance driver safety. Note that in order to evaluate this approach, a touch panel with a secondary task is included in the proposed simulator (see Section 2.2).

Concerning the assistance levels, the shared controller detailed in Section 2.3 is tuned to provide three different torque levels in the steering wheel: gentle, intermediate, and aggressive assistance. Each level is related to a maximum torque level $T_{max}$ that the steering system can apply. The gentle assistance ranges up to 3 Nm (the nominal value defined in the previous section), intermediate up to 6 Nm, and aggressive up to 12 Nm.

The gentle assistance provides a smooth turn that helps the driver react faster, but it is easily overridden by the driver. The intermediate level is the minimum required torque to avoid the crash in a safe way. However, the driver can override this torque level, so in

case the driver does not agree with the correction, the driver can override it. Finally, the aggressive level provides a strong correction torque that overrides the driver. This does not imply that the crash will always be avoided in this mode, as the reaction time is very tight. These three reference torque values were established after iterative testing to identify the most suitable assistance level for each driver state.

The aforementioned six test cases to be studied are listed in Table 2.

**Table 2.** Studied test cases based on driver state and assistance strength.

| Torque Levels | | Driver Awareness | |
|---|---|---|---|
| | | Attentive | Distracted |
| Gentle | <3 Nm | AG | DG |
| Intermediate | <6 Nm | AI | DI |
| Aggressive | <12 Nm | AA | DA |

In addition, to evaluate the performance of the shared control system detailed in Section 2.3, two baseline scenarios will be considered:

1.  **Automation-only baseline**: The automated system performs the evasive maneuver with no driver involvement. This baseline was not executed by the participants.
2.  **Driver-only baseline**: The participant executes the evasive maneuver without any assistance from the automated system.

### 3. Test Procedure

*3.1. Participants*

A total of 24 participants (8 female and 16 male), with an average age of 36 years (ranging from 22 to 59) took part in the study. They were recruited from Tecnalia https://www.tecnalia.com/ (accessed on 2 February 2023) facilities. All participants held university degrees, representing various research fields such as construction, IT, pharmaceuticals, and automotive. Among them, 92% were employed full-time, while the remaining participants were students.

Regarding the driving experience, most of the participants were experienced drivers. Half of them had more than a decade of experience driving and 42% had between 2 and 10 years of experience. Familiarity with driving assistance systems varied; nearly half of the participants had experience with cruise control (CC) or adaptive cruise control (ACC) in their vehicles, although familiarity with lane-keeping systems was limited. A summary of the demographic data can be found in Figures 5 and 6.

All participants signed an informed consent form providing written permission to capture their driving performance data.

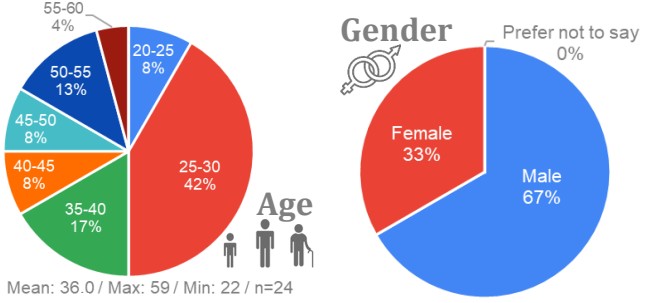

**Figure 5.** Demographic data of the participants: age and gender.

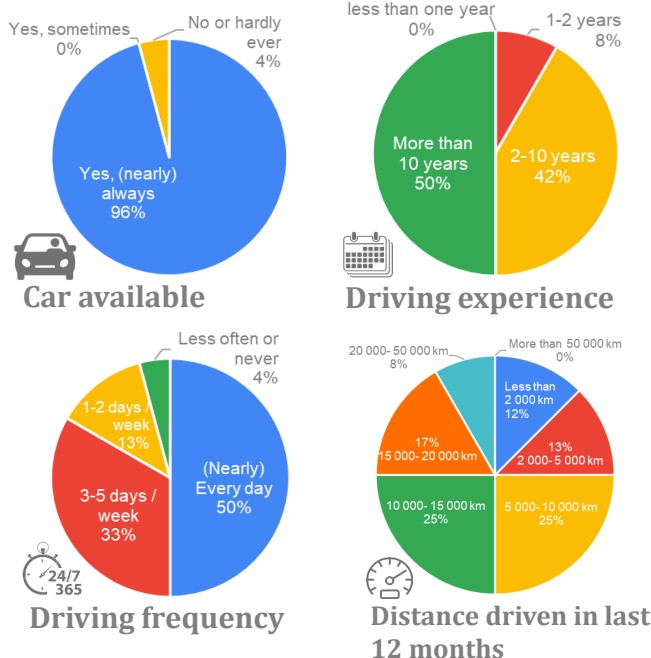

**Figure 6.** Driving experience of the participants.

## 3.2. Test Procedure and Instructions

The participants were instructed to drive the simulator, as detailed in Section 2.2, in the collision prevention scenario outlined in Section 2.1. They completed the scenario seven times, once for each of the different test configurations specified in Table 2, in addition to the driver-only baseline.

As explained in Section 2.1, the scenario incorporated eleven oncoming vehicles, with five motorcycles overtaking in a randomized manner, which participants had to avoid by an evasive lateral maneuver.

Specific instructions provided to the participants varied depending on the particular test configuration. During the attentive tests, participants were asked to keep their eyes on the road and pay attention to incoming events. However, during the distracted tests, participants engaged in a secondary task involving "find the differences" puzzles presented in a strategically placed touch panel within the simulator. This task ensured that the road was out of the participant's field of view while performing it. Participants were instructed to perform the secondary task with their right hand, while keeping the left hand on the steering wheel at all times. This approach ensured that the driver remained partially engaged in the control loop, perceiving the assistance torque provided by the shared control system.

After providing the different instructions, participants were given sufficient time to familiarize themselves with the simulator, allowing them to learn its operation and become comfortable with the driving experience.

## 3.3. Validity and Reliability

The simulator study has been conducted following the guidelines outlined in [30] to ensure a meticulous user acceptance study while minimizing simulator-induced effects. The correlation between a simulator study's validity and an on-road study does not only rely on the fidelity of the simulator. Interestingly, some low-fidelity simulators can provide acceptable validity for specific measures, while some high-fidelity simulators might appear invalid in certain aspects [31].

To hold methodological integrity, a within-subject design was adopted for this study. In this experimental approach, all participants experienced each of the defined configurations. To mitigate any potential bias resulting from the sequence of configurations, the seven test

cases allocated to each participant were evenly distributed across all participants. Table 3 illustrates the equitable distribution of test sequences achieved through the utilization of a Youden square with substitutions. Seven possible orders were established, and each participant followed one of these predefined orders.

Before the beginning of the tests, participants received detailed instructions on simulator operation and the experimental procedures, as previously outlined.

**Table 3.** Test order of each participant for seven cases.

| Participants | Test Order | | | | | |
|---|---|---|---|---|---|---|
| 1 | AI | DI | DG | DA | BL | AA |
| 2 | DI | AA | BL | AG | DG | DA |
| 3 | DG | AG | DA | DI | AI | BL |
| 4 | DA | BL | AG | DI | AA | AI |
| 5 | BL | DA | AI | AA | DG | AG |
| 6 | BL | AA | AI | AG | DI | DG |
| 7 | AG | DG | DI | DG | DA | AI |

**Legend:** AG: Attentive, Gentle. AI: Attentive, Intermediate. AA: Attentive, Aggressive. DG: Distracted, Gentle. DI: Distracted, Intermediate. DA: Distracted, Aggressive. BL: Driver-only Baseline.

To start each session, participants began with manual driving, without any assistance, to reset their familiarity with previously tested systems. Then, participants executed the scenario with one of the test configurations following the predetermined order. After executing each test, participants completed a user acceptance test specific to the configuration tested, rating it in terms of strength and safety on a nine-level Likert scale.

In addition, when performing the different tests, if a crash occurred or the vehicle left the road, a vibration pattern was configured in the steering wheel, so that the participant would be informed of the crash.

### 3.4. Data Collection and Analysis

One of the main contributions of this study is that both quantitative and qualitative metrics were considered. For each of them, statistical analyses based on ANOVA tests were performed. This method is particularly suitable for assessing differences among the means of multiple tests, proving if there is a test that is significantly different from the others.

To address the differences between the six specific configurations, the Tukey HSD post hoc test was conducted. This post hoc test is particularly useful as it allows identifying significant differences of specific configurations in multiple comparisons in a straightforward way, and compared to alternatives like Bonferroni, it provides more precise estimations [32].

### 3.4.1. Quantitative Metrics

Regarding quantitative metrics, three main metrics are used: (1) distance to collision (DTC) between the ego vehicle and the motorcycle, (2) time to collision (TTC), and (3) lateral deviation of the ego vehicle.

**Distance to collision (DTC)** is categorized as follows:

- "Crash" is considered to be when DTC is less than or equal to 0 m.
- "Near Miss" is considered to be when DTC falls between 0 meters and 0.2 m.
- "Safe" is considered to be when DTC is equal to or greater than 0.2 m.

It is important to emphasize why "Near Miss" situations are not regarded as safe. This classification is based on the recognition that a distance of 20 cm between a motorcycle and another vehicle is considered a threshold, considering the typical size of vehicle rear-view mirrors. Such near misses may imply a lateral contact between the two vehicles, posing a safety risk to the motorcyclist.

**Time to collision (TTC)** is measured as the time required for two vehicles to collide if they continue at the present speed and on the same path [33,34]. However, the calculation of TTC can vary depending on the specific requirements and the level of accuracy needed.

The geometric representation of the vehicles can be approximated differently based on the situation. In some instances, vehicles may be treated as points, while in other cases, they are surrounded by a circular envelope considering safety parameters [34,35].

In this study, a more precise approach has been adopted by representing both the vehicle and the motorbike as rectangles. In addition, the rectangle is calculated considering a prediction of the time evolution of both vehicles in the next two seconds for a particular time step. The orientation of each rectangle is calculated with the velocity vector of the vehicle or the motorcycle in each time step. Once the TTC values have been computed, the minimum values of the TTC in each event have been recorded for their comparison.

Finally, the **lateral deviation** is evaluated by determining if the center of the ego vehicle departs from its lane. When this occurs, it is categorized as an "Off-Road" scenario. Considering that the vehicle's width is 2 m, a lateral deviation of up to 1 m on the roadside is deemed acceptable. This decision aligns with the legal requirement in Spain for two-way roads with a speed limit of up to 90 kph, where a minimum roadside width of 1.5 m is required. Therefore, this study adopts a conservative approach in its assessment.

When tallying the number of incidents (crashes, near misses, and off-roads), it is possible for multiple situations to occur simultaneously. In such cases, the following logic is applied:

- If vehicles collide, it is recorded as a "Crash", regardless of any subsequent event.
- If a "Near Miss" occurs without a crash, it is categorized as a "Near Miss".
- If an "Off-Road" scenario is detected, it is classified as an "Off-Road" if neither a "Crash" nor a "Near Miss" has occurred.
- In case none of the previous situations arises, the complete maneuver can be registered as "Safe".

This approach is required as the driving simulator used in the study does not allow simulating post-collision dynamics, making the events immediately following a collision unrealistic for the participant.

### 3.4.2. Qualitative Metrics

Finally, qualitative metrics are used to evaluate user acceptance, safety, and comfort. For that, two questionnaires have been used: (1) a standardized user acceptance questionnaire [36] was used to gauge the perception of usability, usefulness, and overall satisfaction of the participants; (2) a custom questionnaire was used to capture the feelings of strength and safety of the participants. The latter consisted of a nine-level Likert scale to evaluate strength and safety. Finally, participants were also requested to comment on the system to obtain further feedback.

## 4. Results

This section depicts and analyzes the results related to the metrics evaluating the driving performance and user acceptance provided by the participants.

### 4.1. Objective Results

In this subsection, an analysis of the quantitative metrics obtained for the proposed shared control system performance with the participants and the two proposed baselines (automated system only and driver only) will be analyzed and compared.

#### 4.1.1. Baseline 1—Automated System Only

This baseline is used to portray the different levels of assistance provided by the system by evaluating its performance without any driver intervention, which corresponds to the limit condition of the shared control approach (automation only). This way, the shared controller proposed in Section 2.3, with no human intervention, is activated in the motorcycle overtaking scenario in the same conditions as the participants (but with no participant intervention).

It is important to note that the shared control system is not intended (nor allowed) to be used without a human driver, as the system is designed to assist the human driver. However, this test allows us to illustrate the performance of the three different levels of assistance provided by the system (Section 2.4).

Results are shown in Table 4. As can be seen, without the human driver, the automated system consistently evades the oncoming motorcycle at every support level. The intermediate and aggressive assistance levels can fully avoid a collision with the motorbike by executing a complete evasive maneuver and staying on the road. The gentle level, while capable of preventing a collision, experienced a near miss with each motorcycle that crossed its path.

**Table 4.** Crashes, near misses, and off-roads report on the automated system-only baseline.

| Strength | Tests | km | Bikes | Crashes | NM | Off Road | Safe |
|---|---|---|---|---|---|---|---|
| Gentle | 1 | 1.75 | 5 | 0 | 5 | 0 | 0 |
| Intermediate | 1 | 1.75 | 5 | 0 | 0 | 0 | 5 |
| Aggressive | 1 | 1.75 | 5 | 0 | 0 | 0 | 5 |

### 4.1.2. Baseline 2—Driver Only

This baseline is used as a reference for manual driving with no assistance systems, which represents the other limiting condition of shared control. Since no lateral assistance systems were active, none of the participants were able to execute the collision avoidance maneuver safely. Instead, they all ended up departing the lane and losing control of the vehicle.

This outcome suggests that the proposed evasive maneuver was exceptionally challenging to execute. While it cannot be definitively stated that all cases resulted in crashes or lane departures, in every test comprising six safety-critical situations, no driver managed to maintain control of their vehicle throughout.

Furthermore, based on a comparison of their performance with and without the shared control approach, many participants preferred to have some level of assistance to perform the lateral maneuver and stabilize the vehicle. The participants even preferred assistance that felt too strong or uncomfortable.

Since participants were unable to complete this specific test case, obtaining additional data for it was not prioritized. Consequently, no data from this baseline were included in the statistical analysis. Instead, the data serve as a qualitative reference point for discussion.

### 4.1.3. Shared Control System—Driver and Automated System

In this subsection, the main results related to the performance of the participants when facing the evasive maneuver scenario assisted by the different configurations (Section 2.4) of the proposed shared controllers are analyzed. The main performance metrics are summarized in Table 5 for each of the six configurations.

**Table 5.** Crashes, near misses, and off-roads reports of participants in the 6 different test cases.

| | Tests | km | Bikes | Crashes | NM | Off Road | Safe |
|---|---|---|---|---|---|---|---|
| AG | 21 | 34.81 | 102 | 12 | 27 | 6 | 47 |
| AI | 21 | 35.15 | 104 | 16 | 18 | 4 | 66 |
| AA | 21 | 35.02 | 105 | 7 | 17 | 5 | 76 |
| DG | 22 | 38.39 | 110 | 89 | 20 | 0 | 1 |
| DI | 23 | 40.11 | 115 | 34 | 42 | 4 | 35 |
| DA | 23 | 40.11 | 115 | 20 | 33 | 3 | 59 |
| Total | 131 | 223.60 | 651 | 188 | 157 | 22 | 284 |

Please note that due to some unexpected technical malfunctions during the recordings, some datasets needed to be removed. This way, participants drove a total of 223 km in

which 651 overtaking events were recorded successfully. In the recorded data, in 43.63% of the events the motorcycle was safely avoided. This highlights a relatively positive safety performance in the driving sessions, indicating that almost half of the evasive maneuvers were handled safely, which implies an important improvement over Baseline 2, where the participants, without assistance, could not execute safely any test.

Figure 7 summarizes the percentage of each outcome of the evasive maneuvers (crash, near miss, off-road or safe) for the different assistance levels (aggressive, intermediate and gentle) and driver awareness levels (attentive and distracted). It compares these levels with Baseline 1, which represents the performance of the automated system with no driver intervention.

It can be seen that the safety of the system is always improved with respect to Baseline 2 (driver only), but it heavily depends on the assistance level and the driver's awareness level. An aggressive assistance level enables safe execution of the collision avoidance maneuver in 72% of the cases, while distractions reduce it to 51%. The intermediate level reduces success to approximately 63%, still considered acceptable. However, when distracted, this falls to 30%.

Regarding the gentle assistance level, even though it can provide assistance to perform the maneuver safely in 46% of the cases if the driver is attentive, distractions reduce this percentage to only 1%, proving to be almost useless assistance. In any case, the performance of the shared control approach is consistently better than Baseline 2 (driver only), but its performance decreases with respect to Baseline 1 (automated only).

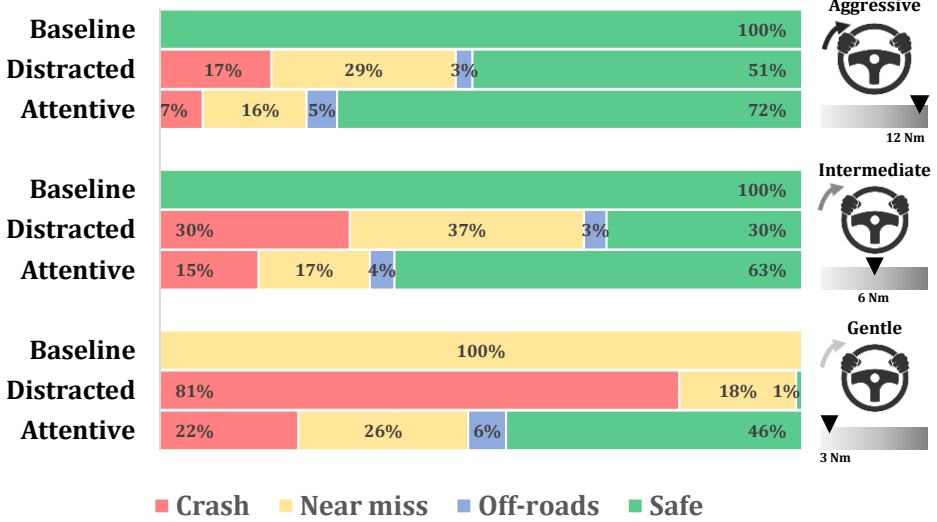

**Figure 7.** Percentage of crashes, near misses, and off-roads for the attentive and distracted drivers, and for the automated-only baseline.

To assess the statistical significance of the data provided, a one-way ANOVA was performed for each of the reported outcomes: crashes, near misses, and off-road incidents. For each of them, the mean of the six studied test configurations (AG, AI, AA, DG, DI, and DA) was compared, resulting in the following F-ratio values. These values represent the degrees of freedom in the analysis: 5 in the numerator (reflecting the number of configurations minus 1) and 15 in the denominator (derived from the total number of observations, 21, minus the number of groups, 6).

- Crashes: ($F_{(5,15)} = 26.94$, $p < 0.05$).
- Near misses: ($F_{(5,15)} = 3.14$, $p < 0.05$).
- Off-roads: ($F_{(5,15)} = 0.96$, $p > 0.05$).
- Safe: ($F_{(5,15)} = 17.06$, $p < 0.05$).

The data related to "Crash", "Near-miss" and "Safe" outcomes show a significant difference in the means of the six test configuration cases ($p < 0.05$). Conversely, the data concerning "Off-Road" incidents revealed no significant difference($p = 0.44 > 0.05$).

To identify the specific groups contributing to this difference, a Tukey's HSD post hoc test was conducted. When analyzing "Safe" cases, a significant difference was observed in comparing attentive and distracted states for both the gentle and intermediate levels:

- DG–AG: $p < 0.05$
- DI–AI: $p < 0.05$
- DA–AA: $p > 0.05$

At the aggressive level, however, the differences are less pronounced, which also can be visually spotted in Figure 7. It is noticeable that as the assistance level increases, the distinction between the two driver awareness states becomes less prominent. This is logical, as the torque provided by the automated system can more easily override the driver as assistance increases.

If unsafe cases are analyzed, the "Crash" scenarios notably decrease with increasing assistance levels. This trend is observed in both attentive and distracted cases, and is especially pronounced in the distracted scenario, where the percentage of crashes drops from 82% in gentle to 17% in aggressive mode. Notably, DG stands out as significantly less safe than the rest of the tested configurations.

Regarding the "Near-miss" incidents, a similar trend emerges, except for the DG case, where a majority of recorded cases are categorized as crashes, reducing the number of "Near-miss" cases.

Figures 8 and 9 show the statistical distribution using boxplots of each tested configuration per resulting scenario, i.e., "Crash", "Near-miss", "Off-road", and "Safe". These graphs confirm the tendencies shown in Figure 7.

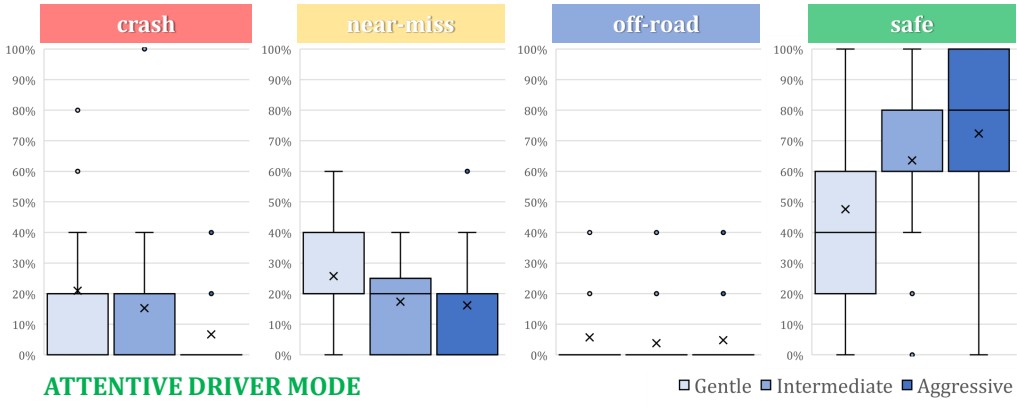

**Figure 8.** Safe cases distribution regarding the 3 assistance levels with an attentive driver.

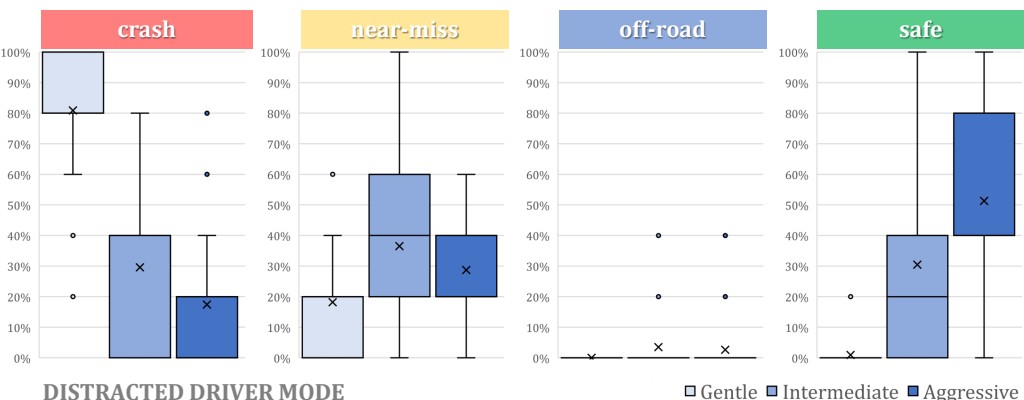

**Figure 9.** Safe cases distribution regarding the 3 assistance levels with a distracted driver.

A more detailed insight into the proposed quantitative metrics is provided in the next figures. Figure 10 shows the distribution of the distance to collision (DTC) for each test configuration (DG, DI, DA, AG, AI, AA) and Baseline 1 (AuG, AUI, and AUA). Note that all configurations related to an attentive driver state exhibit mean DTC values above the "Safe" threshold (>0.2 m). However, in distracted state configurations, only the DA test configuration slightly surpasses this threshold. In DG, only a few outliers avoid crashing.

Figure 11 details the statistical results related to time to collision (TTC). Notably, the DG (distracted driver with gentle assistance) case exhibits a considerably lower TTC compared to the others. Conversely, for the remaining cases, there is a noticeable upward trend in TTC as the level of support assistance increases, especially when the driver is attentive.

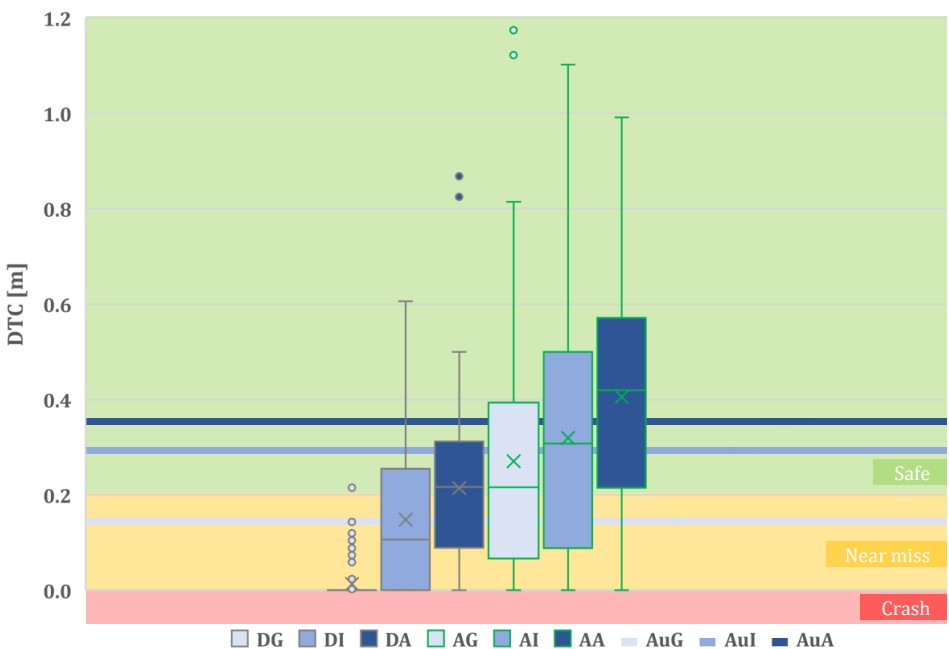

**Figure 10.** Distance-to-collision boxplots for each assistance level and for each driver state and the DTC values obtained for the automated-only baseline.

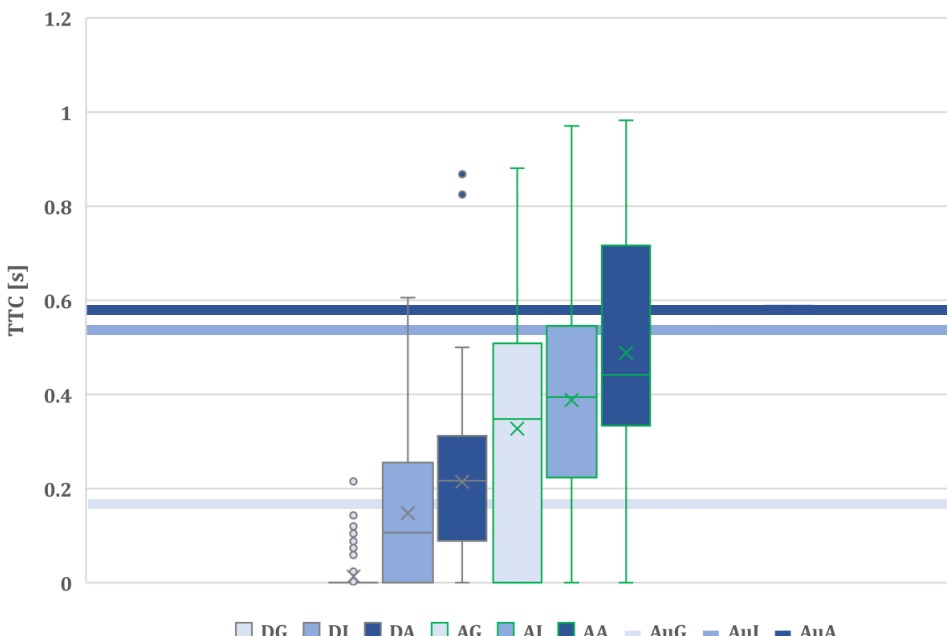

**Figure 11.** Time-to-collision boxplots for each assistance level and for each driver state and the TTC values obtained for the automated-only baseline.

Comparing the performance with the automated system-only test case (Baseline 1), attentive drivers improve the TTC in the gentle mode. However, in all other cases, Baseline 1 consistently shows higher TTC than the driver, regardless of their attentive or distracted state.

In terms of statistical significance, the results of a one-way ANOVA reveal a meaningful difference in the means between the six cases ($p < 0.05$). Subsequently, Tukey's HSD test emphasizes these differences. Specifically, all comparisons between attentive and distracted drivers for each strength value show a significant difference.

- DG–AG: $p < 0.05$
- DI–AI: $p < 0.05$
- DA–AA: $p < 0.05$

This means that an attentive driver state is crucial for successfully avoiding the motorbike, which aligns with expectations.

### 4.2. Subjective Results

As previously detailed, a set of qualitative metrics, based on user acceptance tests and the evaluation of the interaction with the shared control, were employed to perform a subjective evaluation based on the perception of the participants.

Figure 12 presents a comparative analysis of user acceptance scores across three support levels within the attentive and distracted driver modes. The assessment of user satisfaction indicates that the aggressive support mode is the least favored among participants. Similarly, participants perceive the DG mode as dissatisfying due to its limited assistance. In contrast, the intermediate support level receives the highest satisfaction ratings.

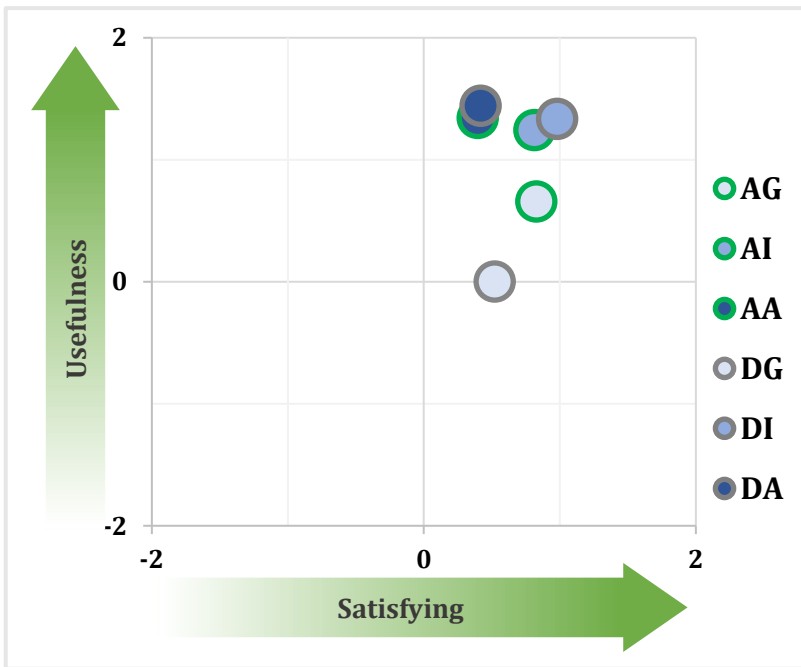

**Figure 12.** User acceptance results regarding attentive (green circle) and distracted drivers (gray circle), with gentle (light blue), intermediate (blue), or aggressive (dark blue) support levels.

Regarding usefulness, both the intermediate and aggressive support levels receive higher scores, while the gentle mode is not considered particularly beneficial. Taking both satisfaction and usefulness into account, the intermediate assistance level emerges as the best option.

Figure 13 illustrates the correlation between perceived strength and safety. Participants perceive the gentle scheme as overly soft and less safe, while the aggressive correction is perceived as excessively strong. This suggests an optimal strength level lying between

the gentle and intermediate levels. Interestingly, all three support levels are perceived as equally safe, except for DG and AA.

In summary, the subjective analysis reveals that neither extreme end of the support spectrum is regarded as user-friendly. Inadequate assistance or excessive intervention is seen as unsuitable. In contrast, the intermediate level reaches a balance between effectiveness and not being overly strong. Overall, it is considered the preferred choice.

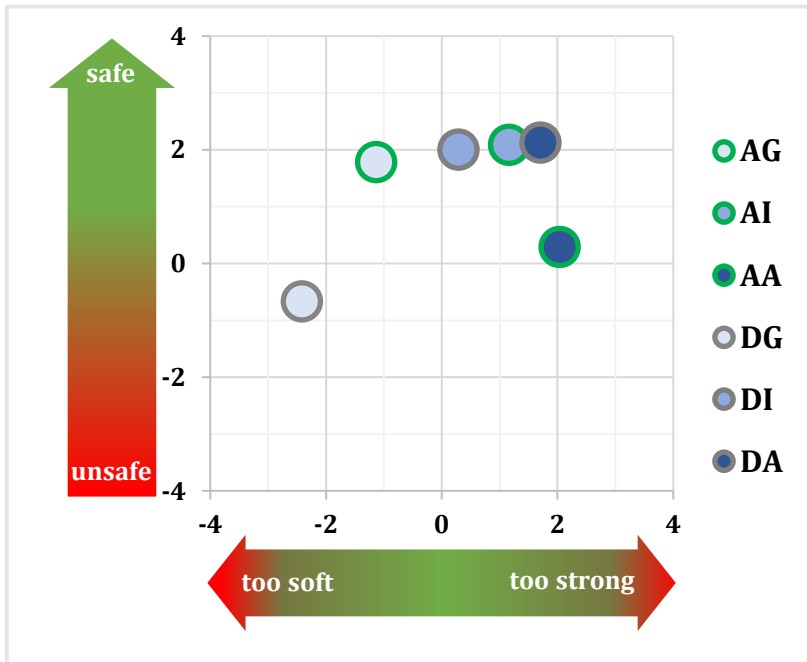

**Figure 13.** Safety–strength results regarding attentive (green circle) and distracted drivers (gray circle), with gentle (light blue), intermediate (blue), or aggressive (dark blue) support levels.

## 5. Discussion

Our study found that the evasive maneuver support functionality effectively reduces the risk and number of accidents, receiving positive user acceptance ratings. Among the three tested schemes, the intermediate level (6 Nm) received the highest subjective evaluation in terms of user acceptance and safety. The subjective ratings regarding the strength level suggest that users would prefer an intermediate or even lower level of strength (3 Nm). However, objective measurements indicate that the aggressive scheme (12 Nm) could potentially further decrease the number of accidents. Nevertheless, user acceptance surveys cannot be neglected, because having a system with low acceptance would mean that it would not be used, which implied results such as the ones shown in Baseline 2 (driver only), with crash scenarios occurring at a frequency of 100% in the proposed critical maneuver. Moreover, even if during the test the participants face up to 21 lateral emergency maneuvers, during the daily use of a vehicle, these critical evasive maneuvers are rare, so it could be considered that the driver would give higher weight to the safety rather than to the acceptance.

On the other hand, the gentle level of assistance should not be considered for this kind of maneuver, as it shows significantly lower metrics. However, it should be remarked that the drivers alone could not perform the maneuver safely during the tests, while having gentle support in attentive mode showed a 46% of success rate, proving that even the worst support system shows a great level of improvement with respect to not using any.

## 6. Conclusions

This paper presents the outcomes of a simulator-based study focused on evaluating a shared control approach designed to execute a safe lateral evasive maneuver, aiming

to prevent collisions. The objective is to provide both objective results related to the performance of the system and subjective evaluations based on user acceptance from participants. These findings offer valuable insights to determine the effectiveness of the shared control approach and its potential implications for real-world implementation.

To perform the study, a set of participants drove a vehicle simulator in which the proposed shared control approach was implemented, and a critical collision avoidance scenario was programmed. A set of six different test configurations were proposed considering the driver state (attentive or distracted), and the assistance level of the controller (gentle, intermediate or aggressive). In addition, two baselines were defined, a driver-only scenario, and an automated system-only one.

Results indicate that although objective measurements suggested that an aggressive assistance level is the safest in terms of the number of accidents, an intermediate level of assistance is the one with higher user acceptance and safety subjective perception. In fact, participants preferred an intermediate or lower level of assistance, prioritizing it over an aggressive one.

Considering the comparison with the proposed baseline approaches, results indicate that shared control approaches, in any configuration, provide a considerable improvement in maneuver safety compared to no assistance at all. However, it should be emphasized that the statistical analysis carried out showed that the gentle level of assistance should not be considered suitable for these maneuvers due to its lower safety performance when distracted.

In summary, the combination of the human driver and the shared control's responsiveness led to superior performance in evasive maneuvers, significantly reducing the risk of accidents with respect to manual driving. However, the automated system itself showed better results working alone rather than in combination with the driver, which shows the complexity of the cooperation and leaves open future works for improvement.

An interesting technology that could emerge to solve the two previous points is a steer-by-wire system, where the feedback received in the steering wheel is not necessarily coupled to the action of the wheels. This way, the steering system in combination with the shared control could perform Baseline 1-like maneuvers, but without the aggressive responsiveness of the steering wheel.

While the study yielded promising results, there are some limitations to acknowledge. The simulation environment, while realistic, may not fully replicate real-world driving conditions. Furthermore, it is based on normal driving conditions, not considering wet or icy roads, which could improve the outcome as they are more limiting driving conditions. Conducting real-world studies with physical vehicles and drivers would complement the results provided in this work and provide a more comprehensive understanding of shared control algorithms' performance and user acceptance. For future works, the identified safer and user-accepted levels of control strength will be applied in a real vehicle.

**Author Contributions:** Conceptualization, methodology, formal analysis, investigation, J.S., M.M., and S.D.; software and validation, J.S. and M.M.; project administration, resources, and supervision, S.D.; review and editing, J.P. and A.Z. All authors have read and agreed to the published version of the manuscript.

**Funding:** This research is supported by the EU Commission HADRIAN project. HADRIAN has received funding from the European Union's Horizon 2020 research and innovation programme under grant agreement No 875597. The publication is supported by the EU Commission Aware2All project, under grant agreement No 97878.

**Informed Consent Statement:** Informed consent was obtained from all subjects involved in the study.

**Data Availability Statement:** Data are contained within the article.

**Conflicts of Interest:** The authors declare no conflict of interest.

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
