# Peer review of "Lateral Evasive Maneuver with Shared Control Algorithm: A Simulator Study"

_sensors, doi:10.3390/s24020562_

Round 1
Reviewer 1 Report
Comments and Suggestions for Authors
This paper presents the results of a simulator-based study to evaluate a shared control algorithm for critical lateral maneuvers. However, the data sources in this paper are narrow and the conclusions drawn lack credibility.
1. The author's selection of driver types is not reasonable, they are all experienced. In addition, considering the differences between drivers, a driver characteristic identification model should be built to cluster different drivers into cautious, general, and aggressive types instead of the two simple divisions of centralized and decentralized in this paper.
2. This paper obtains the data from simulated driving and draws a conclusion, but does not give a reasonable scientific explanation of the conclusion and the improvement of the shared control algorithm. This article is more like an investigation report.
Comments on the Quality of English LanguageEnglish very difficult to understand/incomprehensible
Reviewer 2 Report
Comments and Suggestions for Authors
Overall, the study proposing a shared control algorithm for lateral collision avoidance is well presented and potentially useful for designing automated driving systems. However, the paper needs to be extensively refined prior to publication. Please address the following:
1) The presentation of the results in Section 4 needs to be more professionally organized. For example, the baseline percentages for the gentle setting in Figure 7 are mislabeled and the "unsafe" label in Figure 12 is misaligned.
2) One of the propositions in Section 3.4 says, "Only when a vehicle neither crashes nor has a near miss is it classified as an off-road incident," but this does not account for the safe outcomes also not being near misses or crashes. It would be more logically sound to say, "There is no off-road incident that is a crash or a near miss."
3) Table 3 shows the results for the automation-only scenario, although there is little discussion of how this relates to technologically feasible automated driving systems. It would be more credible to cite previous literature that, if available, coheres with these results and to provide a more detailed technical overview about how these results would be validated through tests with actual automated vehicles.
Most of the sections have acceptable English writing with some minor corrections to be made. For example, the title of Table 2 could be rewritten to say, "Test order of each participant for seven cases."
A bulk of the major language revisions should be focused towards Section 4. The text, figures, and tables stand out from the rest of the manuscript as having numerous language issues. It is recommended that this section be brought up to the same level of writing as the other sections.
Reviewer 3 Report
Comments and Suggestions for Authors
The paper is interesting and well written. Its results further develop authors' previous ones on haptic shared lateral control. I have some minor comments and suggestions:
1. Statistical methods used to assess the results should be briefly introduced and their choice explained (ANOVA, Tukey’s HSD).
2. The results for intermediate assistance in Fig. 7 should be discussed. It seems strange that in this case the portion of safe cases is the same for D and A drivers, however, for aggressive assistance the A drivers reach better number in safe cases than D drivers. Why is the difference between A and D drivers more significant in the aggressive case than in intermediate case? (At least it seems so from Fig.7.)
3. Concerning a previous comment, the 4th paragraph in Conclusion (lines 470-473) seems to describe situation for attentive drivers, not so much for distracted ones?
4. Did the authors consider also maneuvers under different environment conditions (wet or icy road) and the influence of these factors on the results and their dependence on the level of assistance?
Reviewer 4 Report
Comments and Suggestions for Authors
Please see the attachment

1) In line 82, please replace “Unlike” with “In contrast with” (more formal construction)
2) In line 441, please replace “And” with “In addition”
Round 2
Reviewer 1 Report
Comments and Suggestions for Authors
The authors did not fully answer the previously raised comments by the reviewer.
The paper is improved but still very messy and confusing with several technical issues making it unsuitable for this journal.
Comments on the Quality of English LanguageEnglish very difficult to understand/incomprehensible
Reviewer 2 Report
Comments and Suggestions for Authors
The authors have adequately addressed my previous comments.
Comments on the Quality of English LanguageSections of the paper, especially Section 4, show improvement. However, there are still some sentences that need revision. One example is the following sentence in Section 4.
It’s important to note that the shared control approach is not intended (nor allowed) to be used without a human driver.
The informal "it's" should be changed to "it is" to suit a formal publication. Another example in Section 4 is this sentence:
Moreover, after comparing their performance with and without the shared control approach many participants stated preferring having some level of assistance that could help them perform the lateral maneuver and stabilize the vehicle, even if it felt too strong or uncomfortable.
This sentence needs multiple corrections to read as follows:
Furthermore, based on a comparison of their performance with and without the shared control approach, many participants preferred to have some level of assistance to perform the lateral maneuver and to stabilize the vehicle. The participants even preferred assistance that felt too strong or uncomfortable.
It is recommended that the paper be carefully double-checked to ensure that the language is worthy of publication.
